# MGR-Dark: A Large Multimodal Video Dataset and RGB-IR benchmark for Gesture Recognition in Darkness

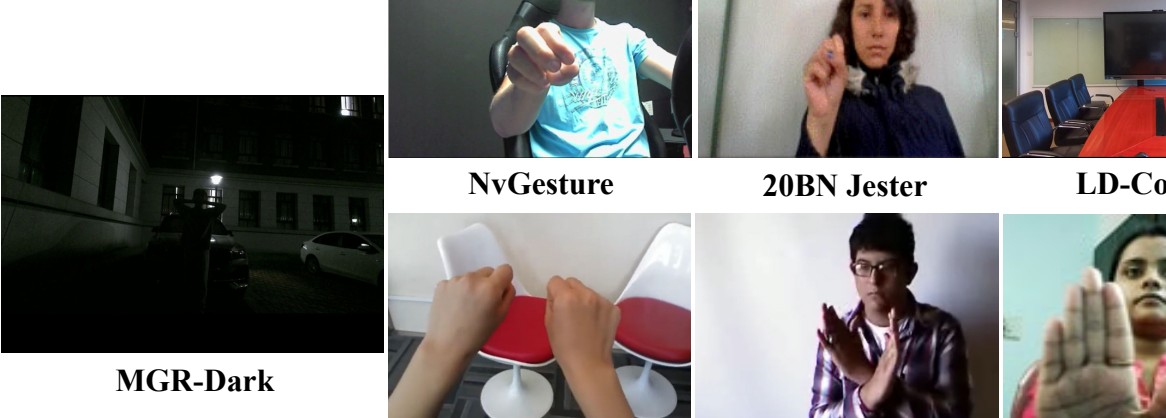

**Figure 1: Example frames from gesture datasets. The bottom of each frame marks the dataset from which it is sampled. The gestures in our dataset MGR-Dark are collected in dark environment and useful in real-life scenes.**

## ABSTRACT

Gesture recognition plays a crucial role in natural human-computer interaction and sign language recognition. Despite considerable progress in normal daylight, research dedicated to gesture recognition in dark environments is scarce. This is partly due to the lack of sufficient datasets for such a task. We bridge the gap of the lack of data for this task by collecting a new dataset: a large-scale multimodal video dataset for gesture recognition in darkness (MGR-Dark). MGR-Dark is distinguished from existing gesture datasets by its gesture collection in darkness, multimodal videos(RGB, Depth, and Infrared), and high video quality. To the best of our knowledge, this is the first multimodal dataset dedicated to human gesture action in dark videos of high quality. Building upon this, we propose a Modality Translation and Cross-modal Distillation (MTCD) RGB-IR benchmark framework. Specifically, the modality translator is firstly utilized to transfer RGB data to pseudo-Infrared data, a progressive cross-modal feature distillation module is then designed to exploit the underlying relations between RGB, pseudo-Infrared and Infrared modalities to guide RGB feature learning. The experiments demonstrate that the dataset and benchmark proposed in this paper are expected to advance research in gesture recognition in dark videos. The dataset and code will be available upon acceptance.

## CCS CONCEPTS

• **Computing methodologies** → **Motion capture**; *Activity recognition and understanding*.

## KEYWORDS

Gesture recognition, MGR-Dark Dataset, Modality Translation, Progressive Cross-modal Distillation

## 1 INTRODUCTION

Gestures play a crucial role in human-computer interaction and language expression, serving as a significant means of information transmission. Numerous datasets[3, 15, 16, 19, 23, 34] and research [4, 17, 24, 33, 36–38] have been dedicated to gesture recognition. However, existing datasets primarily focus on ideal, well-lit environments, neglecting low-light conditions in real-world scenarios. This is partly due to the fact that current benchmark datasets for gesture recognition[3, 16, 23] are normally collected from web videos, which are shot mostly under normal illumination. Yet videos under normal illumination conditions are not available in many cases, such as night surveillance[14, 16], and self-driving at night [4, 19].

In such scenarios, additional sensors like Infrared or Depth imaging sensors can be utilized to recognize gestures in darkness. Furthermore, existing datasets often lack high-quality video data due to limitations in early data collection sensors. With the advancement of sensors, the need for high-quality datasets has also become essential for practical applications.

While multimodal gesture recognition progresses have certainly been made, there are yet many obstacles to be overcome, information transfer between RGB and Infrared poses particular challenges especially. Firstly, the use of different sensors for capturing RGB and Infrared data, along with their differing wide-angle perspectives, leads to horizontal misalignment of images. Secondly, directly using the infrared data stream as the teacher model to guide the student model's RGB data stream may result in the misalignment of feature hierarchies due to the significant modal differences. Many researchers[9, 22, 33] focus on RGB-IR cross-modal distillation. However, it is hard to let the student model simply mimic a pre-trained teacher model when the capacity gap between the teacher and student is large [12, 18, 20]. For instance, Mirzadeh et al.[12] proposed the Teacher-Assistant KD method by gradually increasing the teacher size to foster the distillation process. Inspired by them, we consider that a new intermediate network can bridge the gap between the teacher and student.

In this paper, a large-scale multimodal video dataset (MGR-Dark) is established for gesture recognition in darkness. To the best of our knowledge, it is the first dataset focused on human gesture recognition in the dark. MGR-Dark is distinguished from existing gesture datasets by three highlights. **a). MGR-Dark draws attention to low-light gesture interaction.** Unlike the existing datasets that record gestures in normal light, we capture gestures in a dark environment. Fig. 2 shows example frames sampled from different gesture datasets. It can be seen that the gestures in MGR-Dark are captured with a large field of dark light and are difficult to recognize, which poses a new challenge for gesture recognition. **b). MGR-Dark provides three modalities stream for gesture recognition.** In MGR-Dark, each gesture contains three paired videos,including synchronized RGB, Depth and Infrared stream. It should be noted that we also collected the paired data in daylight(See Figure 3), which provides researchers with more comprehensive data and brings more possibilities for solution gesture recognition in the real world. **c). The videos collected in MGR-Dark are high quality.** The Kinect V2 and XInfrared T3S, equipped with advanced sensors, are used to collect high-quality RGB, Depth, and Infrared video data. The RGB, Depth, and Infrared streams are captured synchronously with resolutions of $1920 \times 1080$, $512 \times 424$, and $1408 \times 1068$ respectively.

To tackle the challenges of cross-modal learning between RGB and infrared data, we propose a novel Modality Translation, Cross-modal Distillation (MTCD) framework which integrates Modality Translation(MT), Progressive Cross-modal Distillation (PCD) module. The two modules address the RGB-IR problem at the image level, and the cross-modality feature level, respectively. First, we employ a modality translator to generate pseudo-Infrared data from the low-light RGB data. Subsequently, in PCD, the teacher network takes the Infrared data as input, the assistant network receives the pseudo-Infrared data as input, while the student model takes the low-light RGB data as input, all of network with features adapted by the attention-based feature converter. During training, the student network(RGB feature learning) is guided separately by the teacher network, the assistant network, and their combination. During inferring, just RGB data in darkness is employed for gesture recognition.

In summary, we explore the task of gesture recognition in dark videos. The contribution of this work is three-fold and is summarized as follows:

• We develop a new Multimodal Gesture Recognition in the Dark (MGR-Dark) dataset, dedicated to the task of recognizing gestures in real-world darkness environment, which, to the best of our knowledge, is the first dataset focused on human gesture recognition in dark videos.

• We propose a Modality Translation and Cross-modal Distillation (MTCD) RGB-IR benchmark framework, exploring pseudo-Infrared stream and progressive cross-modal distillation methods to improve RGB gesture recognition accuracy.

• We benchmark the performance of state-of-the-art gesture recognition models on our dataset and reveal challenges in the task of gesture recognition in dark videos.

## 2 RELATED WORKS

### 2.1 Gesture Recognition Datasets

In the field of gesture recognition, there are several benchmark datasets[3, 15, 16, 19, 23, 34]. For example, the 20BN Jester[16] dataset and the IPN Hand[3] dataset instruct subjects to record gestures using their personal computers or laptops. Subjects sit in front of the computer camera and simulate computer operations using gestures. NvGesture[19] aims to enable gesture-based manipulation of cars, with gestures recorded within a car simulator. EgoGesture[34] concentrates on gesture interaction with wearable devices, collected using Intel RealSense SR300 RGB-D cameras mounted on the subject's head. In the case of ChaLearn IsoGD[23], subjects perform gestures while standing within 1 meter of the Kinect V1 camera under sufficient lighting conditions. The LD-ConGR dataset[15] focuses on long-distance gesture interaction in normal daylight. However, although these datasets encompass a wide range of gestures, it's evident that the gestures are predominantly recorded under normal lighting conditions (See Fig. 1), which limits the capability of current gesture recognition models to videos with normal illumination. In many real-world scenarios, it is necessary to interact with the machine in a low-light environment. Therefore, to investigate gesture recognition performance in dark videos, we have collected a novel multimodal video dataset captured in dark environments.

### 2.2 Cross-modal Knowledge Distillation

Knowledge distillation[11] is a powerful tool for transferring useful information between different domains, such as high-resolution and low-resolution images, RGB and Depth data, etc. There are many cross-modal distillation researches in action recognition. Shi and Kim[21] leverage skeleton data as privileged information to enhance the learning of RNN-based models for action recognition from depth sequences. Bargal et al.[2] design a knowledge distillation framework to learn representations from both depth and RGB videos, while only use RGB data during inference. Similarly, Garcia

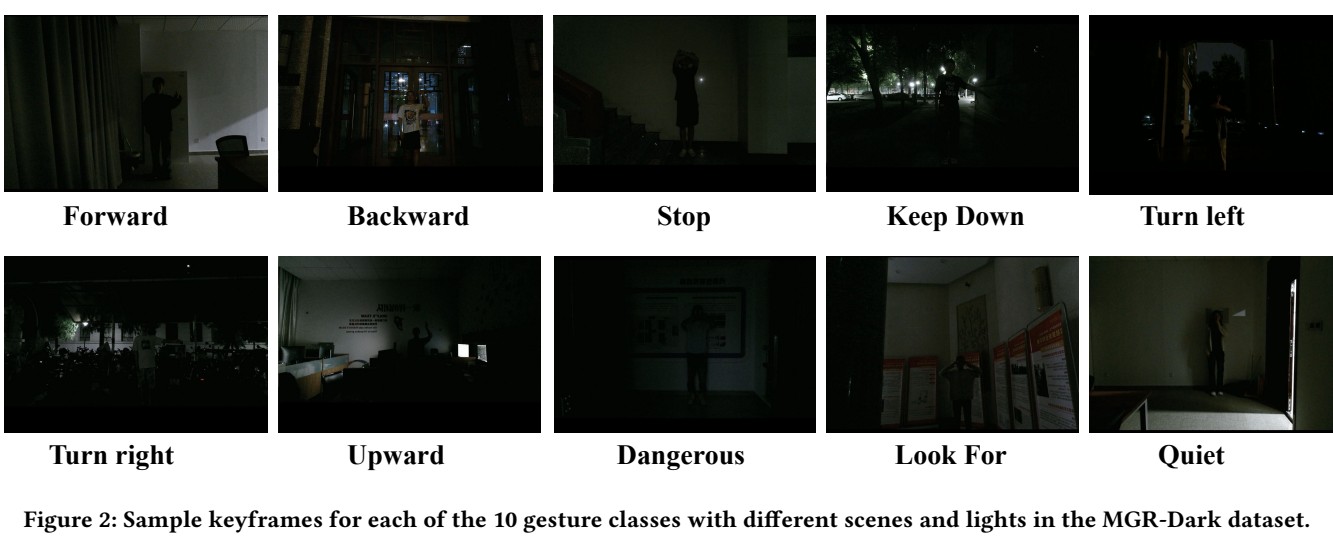

**Figure 2: Sample keyframes for each of the 10 gesture classes with different scenes and lights in the MGR-Dark dataset.**

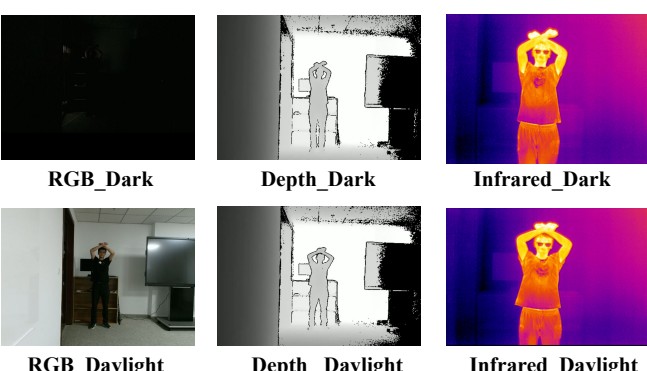

**Figure 3: Sample keyframes for each modality in darkness and the paired samples in daylight.**

et al.[8] trained a hallucination stream, by designing a multi-stage training paradigm and combining an Euclidean-distance metric loss with a generalized distillation loss, for action recognition. However, previous works either ignored the fact that the student modality can contribute useful information to the training of the cross-modal knowledge distillation process or it is not efficient to let the small student network learn directly from a pre-trained teacher network. Our work, instead, utilized both teacher and student modality data to build up assistant (middle)models to discover the relationship between RGB and thermal images. In this way, teacher network will tend to produce a model which is easier for the student network to understand from the human's learning analogy perspective.

## 3 MGR-DARK DATASET

Although a small number of videos taken in the dark do exist in current action recognition benchmark datasets, such as Kinetics[5]and ARID[26], the task of gesture recognition in darkness environments has rarely been studied. This is partly due to the very low proportion of dark videos in current benchmark datasets, and a lack of datasets dedicated to gesture analysis in the dark. To bridge the gap in the lack of dark video data, we introduce a new Multimodal Gesture Recognition in the Dark (MGR-Dark) dataset. In the following subsections, we will introduce the classes, collection and annotation of the dataset, report data statistics, and make a comparative analysis of the MGR-Dark dataset and other gesture recognition datasets.

### 3.1 Data Collection and Annotation

**Collection.** The video clips in the MGR-Dark dataset are collected using Kinect V2 and Xinfrared T3S. We use Kinect V4 to collect RGB-D video data. Kinect V2 is equipped with a 12-megapixel RGB camera and a 1-megapixel Depth camera, ensuring the quality of the captured videos. Xinfrared T3S is a commercial camera equipped with a 12-megapixel Infrared camera. We synchronously record RGB, Depth, and Infrared streams with resolutions of 1920×1080,512× 424, and 1408 × 1068 respectively. The frame rate of 25fps. The MGR-Dark dataset includes a total of 10 gesture classes.The gesture classes include forward, backward, stop, keep down, turn left, turn right, upward, look for, and quiet. We collected videos in 9 outdoor scenes, including carparks, corridors, and groves, and over 17 indoor scenes, such as halls, stairs, and laboratories. The lighting condition of each scene is different, with no direct light shot on the actor in almost all videos. In many cases, it is challenging even for the naked eye to recognize the human gesture without tuning the raw video clips. Figure 2 illustrates sample frames for each of the 10 gesture classes captured in various scenes and lighting conditions in the MGR-Dark dataset. The videos are shot strictly during night hours. In order to expand the application field of the dataset, we further captured paired gesture videos during daylight. Figure 3 shows the samples for each modality in darkness and the paired samples in daylight. The distance from the recording spot to the camera is between 2m and 4m. The videos feature a total of 21 subjects, including 17 males and 4 females. All subjects are shown the standard gestures before recording, and the recorded video will be further checked whether the gestures are correct. The subjects are asked to perform gestures continuously, and a short break is allowed between two gesture instances. The data is only

**Table 1: Comparison of our dataset MGR-Dark and popular gesture recognition datasets.**

| Dataset | Classes | Instances | Light | Distance | Resolution | | |
|---|---|---|---|---|---|---|---|
| | | | | | RGB | Depth | Infrared |
| Jester[16] | 27 | 148,092 | daylight | <1m | *×100 | — | — |
| IPN Hand[3] | 13 | 4218 | daylight | <1m | 640×480 | — | — |
| ChaLearn IsoGD[23] | 249 | 47,933 | daylight | <1m | 320×240 | 320×240 | — |
| NvGesture[19] | 25 | 1532 | daylight | <1m | 320×240 | 320×240 | 320×240 |
| EgoGesture[34] | 83 | 24,161 | daylight | <1m | 640×480 | 640×480 | — |
| LD-ConGR[15] | 10 | 542 | daylight | 1m~4m | 1280×720 | 640×576 | — |
| MGR-Dark | 10 | 31,020 | darkness & daylight | 2m~4m | 1920×1080 | 512×424 | 1408 × 1068 |

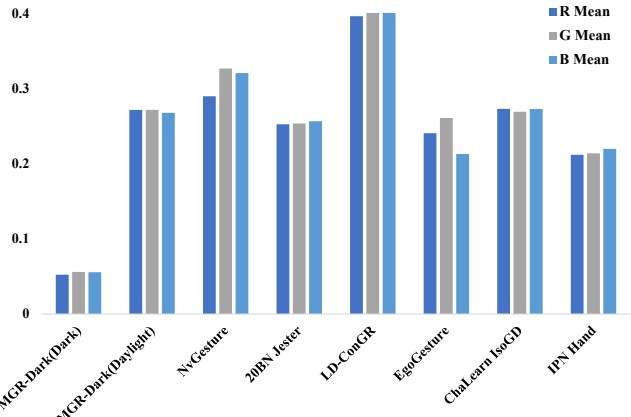 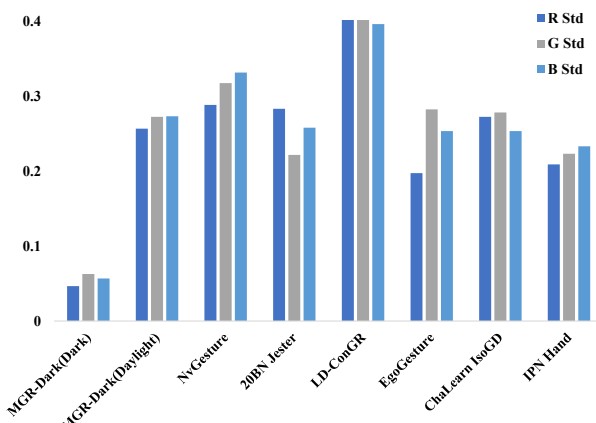

**Figure 4: Bar charts of the RGB mean (left) and standard deviation (right) values for various datasets, including MGR-Dark(dark), MGR-Dark(daylight), NvGesture[19], 20BN Jester[16], LD-ConGR[15], EgoGesture[34], ChaLearn IsoGD[23] and IPN Hand[3]. All values are normalized to the range of [0.0. 1.0]. Best viewed in color.**

allowed for academic research and we will provide strict access for applicants who sign data use agreements. The subjects were informed of the uses of the data and signed informed consent. The gesture label with its video frames can be found in Supplementary.

**Annotation.** In the original dataset, each video sequence contains gestures from 10 different classes, performed sequentially. We manually segmented the data and labeled the temporal segmentation information, indicating the beginning and end frames of each gesture in all MGR-Dark dataset videos. The gestures were synchronized based on time points. Since the RGB and Depth video streams are synchronized, only the RGB streams required labeling, with Depth video annotations obtained accordingly. We utilized automated codes for gesture segmentation based on temporal information, significantly reducing time and labor. Finally, isolated gesture datasets were created.

### 3.2 Data Statistics

The MGR-Dark dataset comprises a total of 31,020 video clips, with 15,510 clips for dark videos and an equal number for daylight videos, due to their paired collection. Each modality, whether dark or daylight, contains 5,170 video clips. With 10 gesture classes, each modality has 517 video clips per class. The dataset was divided randomly into a training set and a testing set based on subjects, with gesture

clips from 16 subjects included in the training set and those from the remaining 5 subjects forming the testing set. All video clips maintain a fixed frame rate of 25fps. Clip lengths range from a minimum of 1.8 seconds (45 frames) to a maximum of 8.4 seconds (210 frames), with the entire dataset spanning 8,721 seconds. Video clips are stored in .mp4 format.

### 3.3 Comparative Analysis

In Tab. 1, we compare our dataset MGR-Dark with the publicly available gesture recognition datasets, including Jester[16]], NvGesture [19], EgoGesture[34], ChaLearn IsoGD[23], LD-ConGR[15] and IPN Hand [3]. Below we will make a detailed comparison and explain the advantages of our dataset from three aspects: light conditions, diversity, and video quality.

**Light conditions.** To better understand the illumination of real dark videos, we compute and compare the statistics of the MGR-Dark(dark) dataset with other datasets. Figure 4 displays the RGB values histograms of datasets MGR-Dark(dark), MGR-Dark(daylight), Jester[16], NvGesture[19], EgoGesture[34], ChaLearn IsoGD[23], LD-ConGR[15] and IPN Hand[3].The histograms of the first two columns represent MGR-Dark, with the first column showing gesture videos in darkness and the second column showing

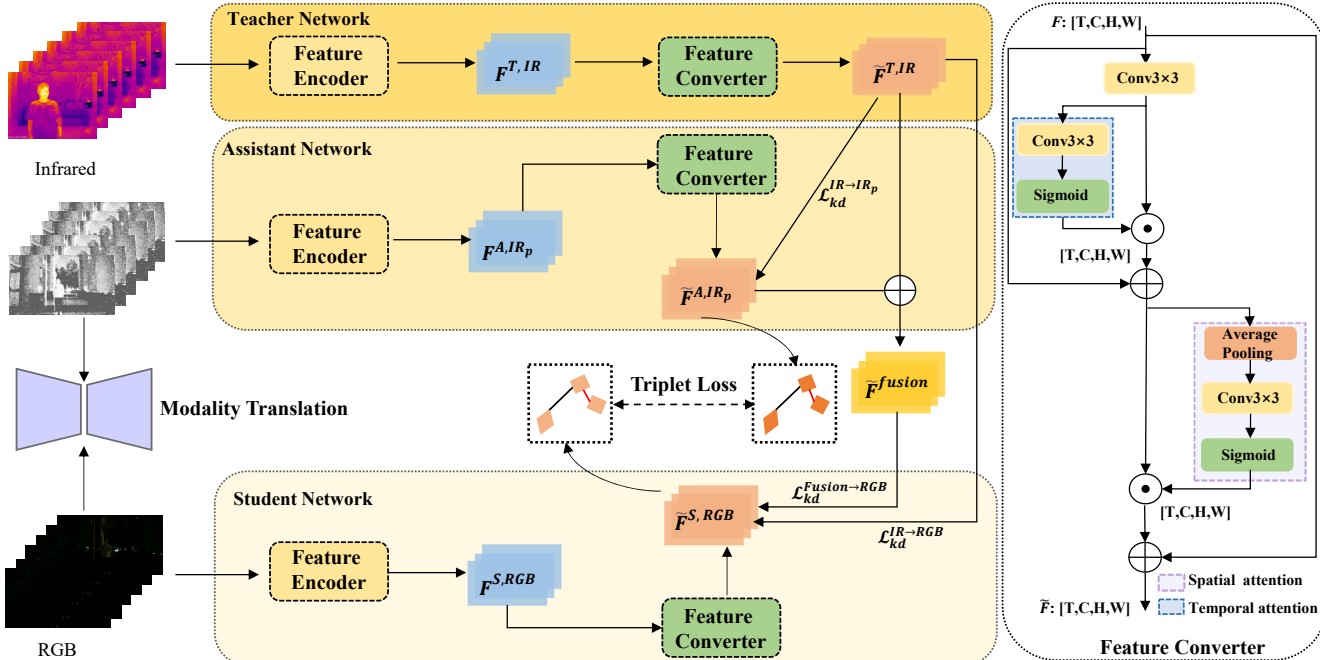

**Figure 5: Overall architecture of our proposed method. The left side is the framework comprising the Modality Translation (MT) and Progressive Cross-modal Distillation (PCD) modules. The right side is Feature Converter. A feature aggregation module with spatial and temporal attention layers.**

paired gesture videos in daylight. The fact that pixels in MGR-Dark(dark), possess lower RGB mean and standard deviation values as shown in Figure 4 implies that video frames in MGR-Dark are lower in brightness and contrast compared to video frames in other datasets. This is further justified by the sampled frames comparison between MGR-Dark(dark) and other datasets, presented in Figure 1. The lower brightness and contrast of video frames in MGR-Dark pose challenges for identifying actors and gestures, even for the naked eye.

**Diversity.** Our dataset covers multimodality and various scenes, and dark gesture videos with paired daylight videos. The videos are collected with three modalities. As we can see in Tab. 2, our dataset provides RGB, Depth and Infrared video data compared to others except NvGesture[19]. Infrared stream is useful for various real-life scenarios, e.g., night surveillance, emergency call for help at night, and driving at night. Compared with NvGesture[19] focus on drive scenes, the videos are collected in more than 25 different scenes, including 9 outdoor scenes and more than 17 indoor scenes, such as carparks, corridors and groves for outdoor scenes, and halls, stairs and laboratories for indoor scenes. To our knowledge, our dataset is the first to provide paired daytime and dark action videos, further enhancing its uniqueness and utility.

**Quality.** As we can see in Tab. 1, our dataset provides high-definition RGB video data ($1920 \times 1080$), while the highest resolution of other gesture datasets is only $640 \times 480$. In addition, the Depth streams (captured synchronously with RGB streams) are available in our dataset and have a higher resolution ($512 \times 424$)

compared to NvGesture[19] ($320 \times 240$) and ChaLearn IsoGD[23] ($320 \times 240$). Meanwhile, the resolution of the Infrared data stream we provide is $1408 \times 1068$.

## 4 METHOD

The overview architecture of the MTCD framework is illustrated in Fig. 5. To tackle the challenges of cross-modal learning between RGB and Infrared data, we propose a novel Modality Translation, Cross-modal Distillation (MTCD) framework, comprising the RGB-to-IR Modality Translation (MT) and Progressive Cross-modal Distillation (PCD) modules. First, we employ a modality translator to generate pseudo-Infrared data from the low-light RGB modality data. Subsequently, we leverage the teacher network and assistant network to integrate the cross-modal knowledge and then guide the feature learning of the student network. The teacher network takes the Infrared data as input, the assistant network inputs the pseudo-Infrared data, and the student network takes the low-light RGB data as input. Three networks with features optimize by the attention-based feature converter. During training, guidance for the student model's RGB feature learning is provided separately by the teacher network, the assistant network, and their combination. During inferring, just the student network with RGB data in darkness is employed for gesture recognition.

### 4.1 Modality Translation

Motivated by recent advancements in Re-Identification[27, 31], we introduce the Modality Translator (MT), based on the domain

translation-based method[9](SDA GAN), to generate pseudo data. Given a source RGB dataset, the MT translates RGB images into Infrared images while preserving the identity of the subjects. This results in a set of original RGB images and their corresponding translated Infrared images. These translated Infrared images serve as pseudo-Infrared data, aiding in guiding cross-model learning. In the experiment, we pretrained the proposed Modality Translation framework on widely used image RGB-IR datasets,SYSUMM01 [25].

Two feature encoders are first pre-trained with two different initialization seeds on SYSUMM01 for providing weight initialization. The encoder-decoder model is a feature transformation function that extracts feature representation $F(x_i \mid \theta)$ from input image $x_i^{RGB}$. The $\theta$ denotes the parameters of the feature encoder network. The training loss can be defined by

$$\mathcal{L}_{pre}(\theta) = \mathcal{L}_{RGB}(\theta) + \lambda^{pre} \mathcal{L}_{Tri}(\theta), \tag{1}$$

where $L_{RGB}(\theta)$ is the cross-entropy loss, and $L_{Tri}(\theta)$ [29] is the triplet loss with hard samples. $\lambda^{pre}$ is the parameter weighting the two losses. $L_{RGB}(\theta)$ and $L_{Tri}(\theta$ [29] are formulated as:

$$\mathcal{L}_{RGB}(\theta) = (x_i, y_i; \theta_{RGB}) = \sum_{k=1}^{K} -p(i) \log(q(i)), \tag{2}$$

$$\mathcal{L}_{Tri} = (x_i; \theta_{RGB}) = \sum_{i,p,n} \lfloor \mathcal{D}_{i,p} - \mathcal{D}_{i,n} + \epsilon \rfloor_+, \tag{3}$$

where $q(i)$ is the predicted label probability and $p(i)$ is the ground-truth label probability, $K$ is the total number of training RGB data in the dataset.$\lfloor x \rfloor_+ = \max(x, 0)$, $\epsilon$ is the margin,$D_{i,p}$ and $D_{i,n}$ denote the L2-norm distance between sample $x_i$ and its hardest positive $x_i, p$ and negative samples $x_i$, $n$ in one batch, respectively, which can be calculated by

$$\begin{aligned} \mathcal{D}_{i,n}(\theta) &= \left\| F(x_i \mid \theta) - F(x_{i,n} \mid \theta) \right\|, \\ \mathcal{D}_{i,p}(\theta) &= \left\| F(x_i \mid \theta) - F(x_{i,p} \mid \theta) \right\|, \end{aligned} \tag{4}$$

where $\| \cdot \|$ is the L2-norm distance between samples.

Unlike traditional translation-based methods[27] focus on homogeneous domains and lack cross-modality supervision, our approach leverages MT to construct cross-modality image pairs, thereby providing essential cross-modality supervision. This helps mitigate the multimodal discrepancy, specifically addressing the misalignment at the image level. Furthermore, the generated pseudo-Infrared data enhances cross-modality learning by enriching the feature space with additional cross-modality information.

## 4.2 Progressive Cross-modal Distillation

Inspired by [20], we introduced a progressive KD learning procedure that requires the student network to learn from the teacher network step-by-step or in incremental method, which can help the student network better learn the information from the teacher network. Specifically, the teacher network takes the Infrared data(IR) as input and fuses, the assistant network receives the pseudo-Infrared(IR$_p$) data as input, while the student network takes the low-light RGB data as input. We leverage the teacher network and assistant network to integrate the cross-modal knowledge and then guide the feature learning of the student network.

As shown in Fig. 5,our Progressive Cross-modal Feature Distillation is divided into three parts: **IR→RGB cross-modal distillation,IR→IR$_p$ cross-modal distillation**, and **Fusion→RGB cross-modal distillation**. In other words, the loss functions of cross-modal feature distillation are formulated as:

$$\mathcal{L}_{kd}^{IR \to RGB} = \frac{1}{2} \left\| (T_t(\widetilde{F}^{T,IR}), T_s(\widetilde{F}^{S,RGB})) \right\|^2, \tag{5}$$

$$\mathcal{L}_{kd}^{IR \to IR_P} = \frac{1}{2} \left\| (T_t(\widetilde{F}^{T,IR}), T_a(\widetilde{F}^{A,IR_p})) \right\|^2, \tag{6}$$

$$\mathcal{L}_{kd}^{Fusion \to RGB} = \frac{1}{2} \left\| (T_a(\widetilde{F}^{Fusion}), T_s(\widetilde{F}^{S,RGB})) \right\|^2, \tag{7}$$

where the feature of the teacher network is denoted as $\widetilde{F}^{T,IR}$, the feature of the assistant network is denoted as $\widetilde{F}^{A,IR_P}$, and the feature of the student network is $\widetilde{F}^{S,RGB}$. To match the feature dimension, $T_t$, $T_a$ and $T_s$ respectively, we transform the feature $\widetilde{F}^{T,IR}$, $\widetilde{F}^{A,IR_p}$ and $\widetilde{F}^{S,RGB}$. A distance d between the transformed features is used as a loss function $L_{kd}^{* \to *}$.

By incorporating the assistant network, the gap between the Infrared and RGB streams are narrowed, facilitating the student model in better mimicking the performance of the teacher network. Specifically, the student model will be updated one step towards the ground-truth labels, guided by the fusion of teacher and assistant features.

## 4.3 Feature Converter

The architecture of our attentive feature converter is depicted on the right side of Fig. 5. Our proposed feature converter takes the initial feature $F$ from the feature extractor as input. It utilizes consecutive spatial attention and temporal attention modules to cooperatively adjust features in both spatial and channel dimensions. This process can be formally described as follows:

$$\widetilde{F} = F + (F + Conv(F) \otimes Att_{spatial}) \otimes Att_{temporal}, \tag{8}$$

where $F : [T, C, H, W]$ signifies the converted feature, while $Att_{spatial}$ and $Att_{temporal}$ represent spatial attention and temporal attention map, respectively. We utilize a $3 \times 3$ convolution layer with sigmoid activation to generate a spatial attention map $Att_{temporal}$ and $Att_{spatial}$, except involving a global average pooling:

$$\begin{aligned} Att_{spatial} &= \sigma(Conv(F')), \\ Att_{temporal} &= \sigma(Conv(avgpool(F'))), \end{aligned} \tag{9}$$

As depicted in Fig. 5, the attention feature converter assumes distinct roles within the Teacher, Assistant and Student networks. In the Teacher and assistant network, the feature converter facilitates the adaptively fused of homogeneous domains in spatial and channel domains, thereby enhancing feature representation learning. In the Student Network, the attentive feature converter mainly focuses on alleviating the feature gap between the RGB feature and the pseudo-Infrared feature with our contrastive feature distillation.

## 4.4 Objective Optimization

In summary, we train the student network and the final total loss for the student network is defined as follows:

$$\mathcal{L}_{total} = \mathcal{L}_{ce} + \alpha \mathcal{L}_{kd}^{Fusion \to RGB} + \beta \mathcal{L}_{kd}^{IR \to RGB}, \tag{10}$$

where $\mathcal{L}_{kd}^{Fusion \rightarrow RGB}$ and $\mathcal{L}_{kd}^{IR \rightarrow RGB}$ means the progressive loss guided by Teacher model and assistant network, $\mathcal{L}_{ce} = \frac{1}{n} \sum_{i=1}^{n} y_i \log \hat{y}_i$ is the standard cross-entropy loss for gesture recognition, and $\alpha$, $\beta$ is corresponding trade-off parameter.

## 5 EXPERIMENTS

In this section, we will first evaluate the state-of-the-art methods in the field of gesture recognition on the MGR-Dark dataset and then discuss the proposed RGB-IR benchmark method on two datasets.

### 5.1 Implementation Detail

All experiments are implemented on the PyTorch with two NVIDIA RTX 3090 GPU. To prepare the input data for training and inference, it is resized to $256 \times 256$ first and then randomly/center cropped into $224 \times 224$ with a length of 32 frames, respectively. During training, the batch size is set to 32, and the training lasts for 80 epochs on all three datasets. The initial learning rate is set to 0.0001, and we decay the learning rate by a factor of 0.1 after 20 epochs. We train all networks using the SGD optimizer. We select 32 frames from each action clip to generate input data, uniformly sampled in time. Additionally, the trade-off parameter $\alpha$ and $\beta$ are 0.9 and 0.3.

### 5.2 Evaluation Metric

To evaluate the performances of the models, we use the recognition rate, $\mathcal{L}_{ce}$, as defined in :

$$\mathcal{L}_{ce} = \frac{1}{n} \sum_{i=1}^{n} f(p(i), y(i)), \qquad (11)$$

where $n$ is the total number of samples, $p$ is the predicted label, $y$ is the groundtruth label; if $p(i) = y(i), f(p(i); y(i)) = 1$, otherwise $f(p(i); y(i)) = 0$.

We refer to this metric as top-1 recognition rate, since we are only evaluating a model's best guess. Top-N recognition rate refers to the rate by which the true class label exists in a model's top-N predictions.

### 5.3 State-of-the-art Evaluation

We evaluate the state-of-the-art gesture and action recognition methods on the proposed MGR-Dark dataset, and the results are presented in Tab. 2.In order to mitigate the influence of different multimodal data processing methods, we conduct separate comparisons based on the RGB or Depth modality.

As shown in Tab. 2, we assess MGR-Dark datset using four different backbones: ResNet[24], 3D CNN[6, 17], 3D ResNet[4], and Transformer [36, 37]. 3DDSN[6] and C3D[17] methods are based on 3D CNN, which simultaneously extract spatial and temporal features via 3D convolutions. In contrast, TSN[24] model processes spatial and temporal information separately by segmenting the video and sampling one frame from each segment. Spatial features are then extracted using 2D CNNs from the sampled frames, while temporal features are represented by optical flow. I3D[4] and our method are based on 3D ResNet. Compared to 3D CNN networks, which can be challenging to train and optimize, the incorporation of residual connections in 3D ResNet reduces the difficulty of training deep networks and enables better capture of spatiotemporal information in video data. The results indicate that the performance

**Table 2: Comparison with SOTAs on MGR-Dark Dataset.**

| Modality | Method | Backbone | Frame | Acc(%) |
|---|---|---|---|---|
| RGB | TSN[24] | ResNet34 | 16 | 63.23 |
| | TSN[24] | | 32 | 65.19 |
| | TSN[24] | ResNet50 | 16 | 65.34 |
| | TSN[24] | | 32 | 68.23 |
| | convLSTM [33] | LSTM | 16 | 60.87 |
| | convLSTM [33] | | 32 | 62.82 |
| | AttentationLSTM[32] | LSTM | 16 | 64.34 |
| | AttentationLSTM[32] | | 32 | 67.44 |
| | gateconvLSTM[38] | LSTM | 16 | 65.23 |
| | gateconvLSTM[38] | | 32 | 68.11 |
| | 3DDSN [6] | 3D CNN | 16 | 54.23 |
| | 3DDSN [6] | | 32 | 57.74 |
| | C3D[17] | 3D CNN | 16 | 49.23 |
| | C3D[17] | | 32 | 50.34 |
| | I3D[4] | 3D ResNet50 | 16 | 75.01 |
| | I3D[4] | | 32 | 77.32 |
| | Decouple&Recouple[37] | Transformer | 16 | 63.12 |
| | Decouple&Recouple[37] | | 32 | 66.32 |
| | UMDR [36] | Transformer | 16 | 65.23 |
| | UMDR [36] | | 32 | 69.22 |
| | **MTCD(Ours)** | 3D ResNet50 | 16 | **82.91** |
| | **MTCD(Ours)** | | 32 | **85.26** |
| Depth | TSN[24] | ResNet34 | 16 | 71.76 |
| | TSN[24] | | 32 | 74.11 |
| | TSN[24] | ResNet50 | 16 | 72.64 |
| | TSN[24] | | 32 | 76.23 |
| | convLSTM [33] | LSTM | 16 | 70.17 |
| | convLSTM [33] | | 32 | 71.82 |
| | AttentationLSTM[32] | LSTM | 16 | 70.82 |
| | AttentationLSTM[32] | | 32 | 72.44 |
| | gateconvLSTM[38] | LSTM | 16 | 74.23 |
| | gateconvLSTM[38] | | 32 | 76.11 |
| | 3DDSN [6] | 3D CNN | 16 | 74.23 |
| | 3DDSN [6] | | 32 | 67.74 |
| | C3D[17] | 3D CNN | 16 | 59.23 |
| | C3D[17] | | 32 | 60.34 |
| | I3D[4] | 3D ResNet50 | 16 | 85.01 |
| | I3D[4] | | 32 | 87.32 |
| | Decouple&Recouple[37] | Transformer | 16 | 73.45 |
| | Decouple&Recouple[37] | | 32 | 75.23 |
| | UMDR [36] | Transformer | 16 | 73.23 |
| | UMDR [36] | | 32 | 75.79 |
| | **MTCD(Ours)** | 3D ResNet50 | 16 | **85.15** |
| | **MTCD(Ours)** | | 32 | **87.45** |

The experiments are mainly implemented on the part of darkness in the MGR-Dark dataset.

of TSN[24] is inferior to that of 3D ResNet-based methods on the MGR-Dark dataset. This is primarily because gestures with long duration may lose keyframes during segmentation and random sampling. Transformer[36, 37] does not perform as well as 3D ResNet due to the vast amount of data required for training transformer networks. Conversely, 3D ResNet demonstrates a strong ability to extract spatiotemporal features. Hence, we focus on 3D ResNet-based models for the backbone of our benchmark. We also notice that though our dataset is of relatively small size and has fewer classes than current normal illumination video datasets, there is plenty of room for improvement in accuracy.

**Table 3: Comparison with SOTAs on NvGesture Dataset.**

| Method | Modality(Training) | Modality(Inference) | Accuary(%) |
|---|---|---|---|
| GPM[10] | RGB,Depth | RGB,Depth | 86.10 |
| PreRNN[28] | RGB,Depth | RGB,Depth | 85.0 |
| Transformer[7] | RGB,Depth | RGB,Depth | 84.60 |
| MTUT[1] | RGB,Depth | RGB,Depth | 85.48 |
| MMTM[13] | RGB,Depth | RGB,Depth | 86.31 |
| NAS[30] | RGB,Depth | RGB,Depth | 83.38 |
| RAAR3DNet[35] | RGB,Depth | RGB,Depth | 88.59 |
| Decouple&Recouple[37] | RGB,Depth | RGB,Depth | 89.75 |
| **MTCD(Ours)** | RGB, IR | RGB | **87.97** |

## 5.4 RGB-IR Benchmark Method

For a comprehensive understanding of our RGB-IR benchmark, we conduct ablation experiments to demonstrate the robustness. We validate the effectiveness of our approach through three aspects: generalization performance, component ablation experiments, and hyperparameter tuning.

**Generalization Evaluation.** To verify the robustness of our model, we test the network utilizing the original with noisy dark video in MGR-Dark without any extra processing. As shown in Tab. 2 and Tab. 4, It is noteworthy that our RGB-IR Benchmark achieves 85.26/87.45% accuracy in RGB and Depth data (higher than all other methods), thus demonstrating its superiority.

Additionally, we further employ NvGesture to test proposed RGB-IR benchmarks. NvGesture Dataset[19] is the only multimodal gesture recognition dataset that provides RGB, Depth, and infrared modalities. As presented in Tab. 3, our method shows its clear advantage against other multimodal methods overall. Although our accuracy is not the highest on this dataset, it still ranks in the top three, which demonstrates that the proposed method can improve the representation of RGB stream by progressive cross-modal distillation, making a better performance. It's worth noting that our method only requires single-modal RGB input during inference.

**The Effectiveness of Components.** For a comprehensive understanding of our model, we conduct ablation experiments to demonstrate the improvements obtained by three important components, including **Teacher network**, **Assistant network**, **Fusion of Teacher and Assistant network**. We ensemble each module to the backbone network step-by-step, and compare the recognition performance on the MGR-Dark. For the convenience of comparison, we construct the baseline that uses the student network with only RGB as input. The results are illustrated in Tab. 4.

A) Teacher network ($\mathcal{L}_{kd}^{IR \to RGB}$). We verify the effectiveness of the Teacher network module first. The results in the first row are obtained from the same depth baseline network, while those in the second row stem from the student network (RGB stream) being distilled by the teacher network (Infrared stream). A comparison of the results in the first and second rows of Tab. 4 reveals that RGB feature learning with the guided by teacher network shows an improvement of nearly 4%. This underscores the crucial role of cross-modal representation in enhancing gesture feature performance.

B) Assistant network ($\mathcal{L}_{kd}^{IR \to RGB}$). With the addition of the Assistant Network (pseudo-Infrared stream), the performance of MTCD further improves. However, although there has been improvement,

**Table 4: Ablation Study of the Key Components in MTCD.**

| Dataset | $\mathcal{L}_{kd}^{IR \to RGB}$ | $\mathcal{L}_{kd}^{IR \to IR_P}$ | $\mathcal{L}_{kd}^{Fusion \to RGB}$ | Acc(%) |
|---|---|---|---|---|
| | × | × | × | 75.58 |
| | ✓ | × | × | 79.15 |
| **MGR-Dark** | ✓ | ✓ | × | 81.13 |
| | ✓ | ✓ | ✓ | 85.26 |

it remains unsatisfactory. This is primarily due to the gap between the RGB and Infrared streams.

C) Fusion of Teacher and Assistant network ($\mathcal{L}_{kd}^{Fusion \to RGB}$). After the fusion of the teacher and assistant network, the results demonstrate that cross-modal representation learning can enhance the progressive cross-modal learning procedure, bringing a prominent performance gain (4.13%). The student network learns from the teacher network in a step-by-step or incremental fashion, thereby helping the student network better mimic the performance of the teacher network. Overall, MTCD significantly outperforms baseline models, underscoring the validity of the three components.

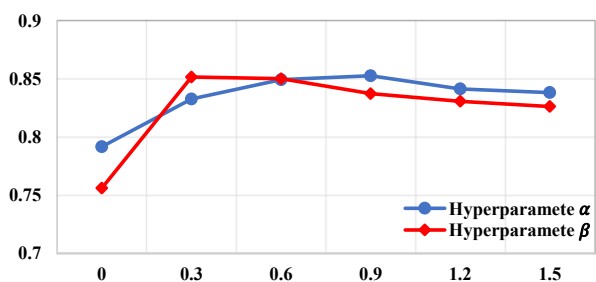

**Figure 6: Effect of the trade-off hyperparameter $\alpha$ and $\beta$.**

**The Effectiveness of hyperparameters.** The parameter $\alpha$ indicates the importance of $\mathcal{L}_{kd}^{Fusion \to RGB}$. We evaluate the scale range setting $\alpha \in [0.0, 1.5]$ as shown in Figure 6. We find that accuracy is improved from 85.26% when $\alpha = 0.9$. As a result, we adopt $\alpha = 0.9$ to achieve the best performance. The parameter indicates the importance of The parameter $\alpha$ indicates the importance of $\mathcal{L}_{kd}^{Fusion \to RGB}$. In Figure 6, we show the influence of the hyperparameter $\beta$. We also evaluate the scale range setting $\beta \in [0.0, 1.5]$. We find that the model achieves the best performance at $\beta = 0.3$, so we set $\beta = 0.3$ as the default in practice.

## 6 CONCLUSION

In this paper, we present a large multimodal video dataset MGR-Dark. MGR-Dark is distinguished from existing gesture datasets by its gesture collection in darkness, multimodal videos (RGB, Depth, and Infrared), and high video quality. To address gesture recognition in darkness, we propose a Modality Translation, Cross-modal Distillation framework. Moreover, representative methods of gesture and action recognition are evaluated and discussed on the MGR-Dark. We believe that our dataset and experimental studies can inspire research in many fields, including but not limited to gesture recognition, action recognition, and human-computer interaction.

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
