# OpenReview forum: "MGR-Dark: A Large Multimodal Video Dataset  and RGB-IR benchmark for Gesture Recognition in Darkness"
_acmmm.org/ACMMM/2024/Conference — MM2024 Poster_

### Official Review · Reviewer_ENdq · 2024-05-15

**Rating:** 4
**Confidence:** 4

**Summary:**

This paper addresses the challenge of gesture recognition in dark environments, an underexplored area crucial for natural human-computer interaction and sign language recognition. The authors introduce MGR-Dark, a large-scale multimodal video dataset designed for gesture recognition in darkness, featuring high-quality RGB, Depth, and Infrared videos. They also propose a Modality Translation and Cross-modal Distillation (MTCD) RGB-IR benchmark framework. This framework includes a modality translator to convert RGB data to pseudo-Infrared data and a cross-modal feature distillation module to enhance RGB feature learning using Infrared data. The experiments show that this dataset and benchmark framework significantly advance research in dark environment gesture recognition.

**Strengths:**

The strengths of this paper are three-fold. Firstly, it introduces the MGR-Dark dataset, specifically designed for gesture recognition in dark environments, which fills a notable gap in existing research. Secondly, it develops a Modality Translation and Cross-modal Distillation (MTCD) RGB-IR benchmark framework that uses pseudo-Infrared data and progressive cross-modal distillation to enhance RGB gesture recognition accuracy. Lastly, the paper benchmarks the performance of state-of-the-art gesture recognition models on the MGR-Dark dataset, revealing specific challenges in recognizing gestures in dark videos.

**Limitations:**

1. The paper contains numerous writing issues that significantly reduce its quality.

a). There are many instances where necessary spaces are missing in the abstract and main text. For example, in the abstract: "multimodal videos(RGB, Depth, and Infrared)"; in the Introduction: "the paired data in daylight(See Figure 3),", "the student network(RGB feature learning)"; in the Method section: "As shown in Fig. 5,our Progressive"; in the experiments: "presented in Tab. 2.In order to mitigate"; and in several places in the references, like “Numerous datasets[3, 15, 16, 19, 23, 34]”、“recognition[3, 16, 23]” et al. The authors should carefully check the entire manuscript.
b). There are inconsistencies in the figure citation format, such as "Fig. 2, Fig. 5" versus "Figure 2, Figure 3, Figure 4". The authors should ensure consistent citation formatting throughout the paper.
c). There is an incorrect table citation in section 3.3 Diversity: "As we can see in Tab. 2, our...". The authors should thoroughly check for such errors.
d). References should be added to the dataset in Figure 1.

2. In Section 3.1, there is confusion regarding the equipment used for data collection: "The video clips in the MGR-Dark dataset are collected using Kinect V2 and Xinfrared T3S. We use Kinect V4 to collect RGBD video data. Kinect V2 is equipped with a 12-megapixel RGB camera and a 1-megapixel Depth camera, ensuring the quality of the captured videos." It is unclear whether Kinect V2 or Kinect V4 was used.  Additionally, the setup and positioning of Kinect and Xinfrared devices during data collection should be illustrated (use a graph or table to illustrate), including any angular differences and whether all collection environments were the same.

3. The information shown in Figure 4 seems unnecessary for the paper's quality. Calculating the mean and variance of grayscale images would suffice. It is recommended to integrate this information with Table 1.

4. In the Feature Converter Module, the rationale behind the sequence of Temporal attention followed by Spatial attention needs clarification. Does the order have any significant impact on the experimental results?

5. There are unfair comparisons in Table 2. For instance, the methods Decouple&Recouple [37] and UMDR [36] are originally based on RGB-D recognition, but the comparisons are made using only RGB or Depth, whereas the authors’ method uses RGB+Infrared during training. Furthermore, the process for obtaining the Depth results in Table 2 is unclear. Considering that Figure 5 describes a process from RGB to pseudo-Infrared, does a similar process exist for Depth? Or is the model trained based on the method described in Figure 5 and then tested directly using Depth?

**Suitability:**

3

---

### Official Review · Reviewer_LLTs · 2024-05-23

**Rating:** 3
**Confidence:** 4

**Summary:**

This paper introduces MGR-Dark as a multimodal gesture recognition dataset, where part of the data is collected under dark conditions.

**Strengths:**

The dataset is indeed interesting, expanding the gesture recognition data towards adverse environments, yet there are multiple concerns that the authors should address.

**Limitations:**

**1. Selection of class is unclear.** How are the 10 classes selected? MGR-Dark seems to have the smallest number of classes compared to other gesture recognition datasets. Why are these particular classes selected or collected? How are these classes characterized?

**2. Detailed statistics are missing.** Several key statistics are missing for the dataset and benchmark. For example, what is the train:test ratio of the dataset? Does the testing set include only dark videos or it also include daytime videos? How many videos are there for each video length (since the video length span from 45 frames to 210 frames, where the longest videos is almost 5 times the length of the shortest videos)? Also how many videos are there for each different depth? What are the specific scenes and why are these specific scenes selected?

**3. The dataset risks to be biased towards gestures performed by certain groups of people.** From the provided statistics and screenshot, we observe that the dataset is collected from a group of volunteers who are mainly (if not all) Asians and mostly male (male:female ratio over 4:1). If the dataset is biased towards gestures performed by such an ethnicity and gender, models trained on this dataset would not be generalizable towards the more general gesture recognition, and the dataset would prove rather useless.

Some further but rather minor concerns are as follows:

a. While the authors stressed on the multi-modality of the dataset, it should be noted that in real-world scenarios, it is common to use only the RGB modality due to the low cost of ordinary cameras. Is the trained model able to generalized to RGB-only scenarios?

b. The authors have not provided how the mosaic of the person filmed would be formed in the example provided, leading to doubt about whether the mosaic can protect the privacy of the person filmed.

**Suitability:**

3

---

### Official Review · Reviewer_kDxy · 2024-05-26

**Rating:** 5
**Confidence:** 4

**Summary:**

This paper proposes a new large-scale multimodal dynamic gesture recognition database based on dark environments. Further, a distillation benchmark network is introduced using the database.

**Strengths:**

•	Gesture recognition is important for various applications. This paper presents a gesture recognition database collected in the dark environments. According to the author, this is among the first work to help the community in night surveillance and self-driving fields.
•	The database is high-quality and provides high-resolution images, which are very useful in improving accuracy.
•	The experiments are well conducted.

**Limitations:**

•	This brightness, contrast and pixel intensity can be manipulated using the image processing techniques. Further, recent GAN-based methods can also generate the dark samples using the existing datasets.
•	If we use simulated images, then what is the performance using real (proposed dataset) and simulated datasets. It is needed to be verified. Further analysis among the RGB, Depth and Infrared images should be conducted to provide the characteristics difference between images captured in dark and simulated.
•	In Figure 5, should be re-drawn to clearly depict the information flow. The output of Modality Translation is not provided to other modules. Both arrows are shown as input to Modality Translation.

**Suitability:**

2

---

### Official Review · Reviewer_YjnS · 2024-05-29

**Rating:** 4
**Confidence:** 2

**Summary:**

The paper presents a significant contribution to the field of gesture recognition with the introduction of the MGR-Dark dataset and the MTCD framework. The research is well-conducted, and the results are promising. However, there is room for expansion and further exploration of the framework's limitations and potential applications.

**Strengths:**

1. The introduction of a new dataset that fills a gap in the current research landscape.
2. The innovative approach to cross-modal learning between RGB and Infrared data.
3. Comprehensive experiments and ablation studies that validate the proposed framework.

**Limitations:**

1. While the dataset is large, the number of gesture classes is limited, which might affect the generalizability of the results.
2. The paper could benefit from a broader discussion on the potential limitations of the MTCD framework and how it might be adapted or extended in future work.
3. Given the high resolution of the dataset, I am curious about the efficiency of the methods in processing this data.

**Suitability:**

2

---

### Meta-Review · Area_Chair_p5Jq · 2024-07-04

**Recommendation:** Accept (Poster)
**Confidence:** 4

**Metareview:**

The paper proposes a new multimodal dataset that fills a rather unexplored area of study $-$ gesture recognition in the dark. This should benefit the current research community with all reviewers equally receptive to this proposal. The paper also goes to the extent of providing competitive baselines and introducing a new MTCD framework that aids the transfer (or distillation) of information and relations between the modalities. I project that this dataset would advance the current research in gesture recognition, especially since the possibility of adverse capture conditions can hinder well-performing methods. There are some presentational / writing issues in the paper, which were rightly pointed out by one reviewer. Also, some sections of the paper that lack clarity (as already answered in the rebuttal) should be properly addressed in the final version to increase the quality of the paper. There were some minor concerns regarding the proposed MTCD method but I think the paper has placed a tremendous emphasis on the initiation of the dataset, and hence it would be slightly unreasonable to overcrowd the paper with too many details. All things considered, I can recommend that this paper be accepted as a poster at ACM MM.